# The Combination of Modified Mitchell’s Osteotomy and Shortening Oblique Osteotomy for Patients with Rheumatoid Arthritis: An Analysis of Changes in Plantar Pressure Distribution

**DOI:** 10.3390/ijerph18199948

**Published:** 2021-09-22

**Authors:** Hyunho Lee, Hajime Ishikawa, Tatsuaki Shibuya, Chinatsu Takai, Tetsuya Nemoto, Yumi Nomura, Asami Abe, Hiroshi Otani, Satoshi Ito, Kiyoshi Nakazono, Kaoru Abe, Kazuyoshi Nakanishi, Akira Murasawa

**Affiliations:** 1Department of Rheumatology, Niigata Rheumatic Center, Niigata 957-0054, Japan; med.racenter@gmail.com (H.I.); rareha@tuba.ocn.ne.jp (T.S.); chinatsunachi.89@gmail.com (C.T.); tetsuya.nemoto.md@gmail.com (T.N.); ynomu@med.kagawa-u.ac.jp (Y.N.); asami.abe@nifty.com (A.A.); fwks9963@nifty.com (H.O.); s-ito@water.ocn.ne.jp (S.I.); wtsgs483@ybb.ne.jp (K.N.); akirakku.m@gmail.com (A.M.); 2Department of Orthopaedic Surgery, Nihon University School of Medicine, Tokyo 173-8610, Japan; nakanishi.kazuyoshi@nihon-u.ac.jp; 3Department of Prosthetics, Orthotics and Assistive Technologies, Niigata University of Health and Welfare, Niigata 950-3198, Japan; kao-abe@nuhw.ac.jp

**Keywords:** forefoot deformity, joint-preserving surgery, rheumatoid arthritis, plantar pressure

## Abstract

The present study aims to evaluate changes in plantar pressure distribution after joint-preserving surgery for rheumatoid forefoot deformity. A retrospective study was performed on 26 feet of 23 patients with rheumatoid arthritis (RA) who underwent the following surgical combination: modified Mitchell’s osteotomy (mMO) of the first metatarsal and shortening oblique osteotomy of the lateral four metatarsals. Plantar pressure distribution and clinical background parameters were evaluated preoperatively and one year postoperatively. A comparison of preoperative and postoperative values indicated a significant improvement in the visual analog scale, Japanese Society for Surgery of the Foot scale, and radiographic parameters, such as the hallux valgus angle. A significant increase in peak pressure was observed at the first metatarsophalangeal joint (MTPJ) (0.045 vs. 0.082 kg/cm^2^; *p* < 0.05) and a significant decrease at the second and third MTPJs (0.081 vs. 0.048 kg/cm^2^; *p* < 0.05, 0.097 vs. 0.054 kg/cm^2^; *p* < 0.05). While overloading at the lateral metatarsal heads following mMO has been reported in previous studies, no increase in peak pressure at the lateral MTPJs was observed in our study. The results of our study show that this surgical combination can be an effective and beneficial surgical combination for RA patients with mild to moderate joint deformity.

## 1. Introduction

Patients with rheumatoid arthritis (RA) are susceptible to a variety of foot problems that result in the limitation of activity in their daily lives [1,2,3]. In the progression of rheumatoid arthritis and forefoot deformities, such as subluxation or dislocation of the metatarsophalangeal joint (MTPJ) in the lesser toe, hallux valgus, and spread foot, often occur due to a decrease in the transverse foot arch, and painful callosities are frequently formed in the forefoot. Foot health is clearly a major factor in the quality of life of RA patients [4] and should be a primary consideration in the selection of appropriate treatment. In RA patients with painful callosities, treatment methods such as foot orthoses and foot care have been attempted. It was reported that the wearing of foot orthoses resulted in a mean reduction in plantar pressure at forefoot regions of interest of 9% [5]. Conversely, it was reported that the debridement of painful forefoot plantar callosities produced no additional benefits in the long term [6]. Accordingly, forefoot reconstructive surgery has become a more popular option for RA patients with painful callosities [7,8,9]. In addition, Shimoda et al. reported that forefoot arthroplasty had no adverse effects on gait parameters and no connection to disability [10].

In a previous study, we reported a significant correlation between plantar pressure, dislocation, and callosity, and that dislocation or callosity could be quantitated by plantar pressure measurement [11]. Konings-Pijnappels et al. also concluded that forefoot deformity is related to high plantar pressure [12]. These studies seem to indicate that plantar pressure at a callosity site would decrease following reconstructive surgery.

Several studies have reported on plantar pressure in patients with RA. In a study comparing RA patients to a control population, load bearing was seen to be smaller at the first toe and the first metatarsal head and greater at the third through fifth metatarsal heads [13]. It has also been reported that peak pressure at the rearfoot and pressure time integrals at the forefoot and rearfoot in patients with RA were higher when compared to a control population [14]. In addition, several studies concerning forefoot reconstructive surgery in RA patients have reported on changes in plantar pressure distribution. A significant reduction of plantar pressure at the second and third metatarsal head area during walking and standing in RA patients was observed following head resection of the first through fifth metatarsals [15]. Meanwhile, a lateral shift in plantar pressure distribution was reported to result following a procedure in which a silicon joint replacement was performed for the first MTPJ and a metatarsal head resection was performed for the second through fifth MTPJs [16]. A previous study of ours evaluated the changes in plantar pressure distribution resulting from the following surgical combination: silicon joint replacement performed for the first MTPJ and shortening oblique osteotomy (SOO) performed for the second through fifth metatarsal necks. The results of that study revealed a significant decrease in peak pressure at the second and third MTPJs, and a significant decrease in peak pressure at the first interphalangeal joint (IPJ) [17]. This decrease at the first IPJ is reflected in the lack of force generated during the extension of the first toe, which may present a problem for RA patients with a high degree of activity in daily living. Yano et al. reported a joint-preserving surgery with the following combination: proximal rotational closing-wedge osteotomy performed for the first metatarsal and SOO performed for the second through fifth metatarsal necks. A significant extension in movement distance at the center of pressure was observed following this combination [18]. As can be seen from the studies mentioned above, there are several types of rheumatoid forefoot surgery. Horita et al. reported that a recurrence of callosities and claw toe deformity after surgery was observed more frequently in a resection arthroplasty group than in a joint-preserving arthroplasty group [19]. Ebina et al. also reported that patient-based outcomes in a joint-preserving surgery group were better than those in a resection-replacement group in terms of plantar pressure distribution [16].

Good clinical results were reported in a previous study of ours using a joint-preserving surgery, which consisted of modified Mitchell’s osteotomy (mMO) of the first metatarsal neck and SOO of the second through fifth metatarsal necks (mMO and SOO) [20]. The purpose of the present study is to investigate the changes in plantar pressure distribution in RA patients using this same combination of surgical procedures. To the best of our knowledge, this is the first study to attempt such an evaluation.

## 2. Materials and Methods

### 2.1. Study Design and Participants

A retrospective study was conducted in RA patients who had undergone a surgical combination consisting of an mMO and an SOO at our hospital from May 2012 to September 2017. Surgical treatment was offered to patients who had a hallux valgus deformity and a lesser toe deformity with persistent painful callosities in the forefoot as an alternative to conservative foot care such as foot orthoses and callosity shaving. Plantar pressure measurement was performed preoperatively and one year postoperatively. Data were collected and analyzed retrospectively. The exclusion criteria included a history of previous foot surgery and severe hip, knee, or ankle joint deformity equal to, or higher than, Larsen grade III, under the assumption that plantar pressure would be influenced by malalignment of the lower limbs. Twenty-six feet in 23 RA patients, consisting of 24 feet in 21 female patients and two feet in two male patients, were included in the present study. Patient characteristics are shown in Table 1. The median age was 60.0 years (interquartile range: IR 48.0–70.0), the median duration of RA was 11.0 years (IR 7.0–15.0), and the median body mass index was 21.5 kg/m^2^ (19.9–23.3). Larsen grade for the first MTPJ in the present study was grade II for 16 feet and grade III for 10 feet. This study was conducted according to the guidelines of the Declaration of Helsinki and approved by the ethics committee of our institution (approval number: 2017–011). All patients in the present study gave their informed consent for participation and for the publication of their anonymized data.

### 2.2. Surgical Procedures

The surgical procedure for mMO used in the present study has been reported previously [21,22,23] and was performed as follows: An arced dorsomedial incision was made at the first MTPJ, and a distally based capsular flap was created by making a Y-shaped capsular incision over the medial aspect of the first MTPJ. A lateral release of the first MTPJ was also performed. The distal incomplete cut of the double osteotomy was made at a level one finger in breadth removed from the first metatarsal end. The proximal complete cut of the double osteotomy was made at a level 7 to 10 mm further proximally. The lateral step was shaped by a subsequent perpendicular cut at the distal margin. Following this, the metatarsal head was supinated and shifted laterally, while fitting the proximal lateral margin to the distal lateral step. The osteotomized section was generally stabilized by a Kirschner wire and a soft wire.

The surgical procedure for SOO used in the present study has also been reported previously [20] and was performed on the lateral metatarsals according with the following technique: A dorsal incision was made between the second and third distal metatarsals and between the fourth and fifth distal metatarsals. The extensor digitorum longus was preserved, and the extensor digitorum brevis was severed. However, if severe MTPJ dislocation was detected, the extensor digitorum longus was separated using Z-lengthening. The first oblique bone incision was made at the metatarsal neck at a slope of 45 degrees. The second oblique bone incision was made 7 to 10 mm proximally, parallel to the first osteotomy. The plantar osteophyte of the metatarsal head was often resected. A lesser toe deformity was manipulated gently, with care given to avoid blood vessel injuries. However, if the deformity was rigid and manual correction could not be achieved, resection of the proximal phalangeal head was performed. Following this, the osteotomized section was stabilized by a Kirschner wire, which was applied from the distal phalanx to the proximal metatarsal, along the slope of the distal stump.

Heel contact was allowed from the day following the surgery. The Kirschner wires used to shorten the oblique osteotomy at the second through fifth metatarsal necks were removed three weeks after surgery, after which full weight bearing was allowed using an insole with transverse and longitudinal arch support. The insole was used until three months after surgery.

### 2.3. Patient Background Parameters

Values for the following parameters were obtained before surgery and one year after surgery: methotrexate (MTX) usage, biologics usage, prednisolone (PSL) usage, pain visual analog scale (pain VAS) [24], general visual analog scale (general VAS) [24,25], disease activity measured using a disease activity score in 28 joints based on erythrocyte sedimentation rate (DAS28-ESR) and a simplified disease activity index (SDAI), daily movement using a Japanese version of the Stanford health assessment questionnaire (J-HAQ), and foot function using the Japanese Society for Surgery of the Foot RA foot and ankle scale (JSSF RA scale) [26,27]. In addition, the following angles were measured using a standing anteroposterior radiograph: hallux valgus angle, the angle comprising the first metatarsal and the second metatarsal, and the angle comprising the first metatarsal and the fifth metatarsal. Patient background parameters are shown in Table 2.

### 2.4. Plantar Pressure Measurement

An F-SCAN II system (Nitta Corporation, Osaka, Japan) was used to measure plantar pressure distribution using peak pressure values. A sensor-sheet was applied to the patient’s bare foot. Plantar pressure was measured while patients walked according to their accustomed pace and gait. The value at the third step, which provides an accurate representation of normal gait according to the mid gate method, was recorded [28,29]. As in our previous studies, the plantar pressure distribution image was overlayed on the radiographic imaging, and the following nine sections were identified to calculate the peak pressure at each section: the first IPJ, the first, second, third, fourth and fifth MTPJs, the medial midfoot, the lateral midfoot, and the hindfoot [11,17] (Figure 1). The long axis line between the second and third MTPJs was used to determine the medial and lateral midfoot sections. Plantar pressure values at these nine sections were obtained preoperatively and one year postoperatively and then compared. Additionally, the percentages in which a maximum peak pressure value or minimum peak pressure value occurred in each section were evaluated preoperatively and one year postoperatively. Maximum and minimum peak pressure indicates the highest and lowest peak pressure value found at each of the nine sections. The peak pressure values for each subject were divided by the subject’s body weight in order to calculate peak pressure per 1 kg body weight.

### 2.5. Statistical Analysis

Data analysis was performed using IBM SPSS software, version 25. A Fisher’s exact test was used to investigate the differences in medication usage ratios preoperatively and one year postoperatively. Differences between preoperative and postoperative values were evaluated using a Wilcoxon’s signed rank test. P values of less than 0.05 were considered significant.

## 3. Results

No significant differences in MTX, biologics, and PSL usage ratios were detected in the values recorded preoperatively and one year postoperatively (Table 2). The median doses were from 8.0 mg/week (IR 8.0–10.0) to 8.0 mg/week (IR 8.0–11.0) for MTX, and from 3.0 mg/day (IR 2.0–4.0) to 3.0 mg/day (IR 2.0–3.0) for PSL. Breakdown of preoperative biologics usage was infliximab for one patient, adalimumab for one patient, golimumab for two patients, tocilizumab for two patients, and abatacept for three patients. In addition, one more patient received adalimumab, and one patient received certolizumab pegol one year after surgery. Pain VAS (31.0 mm vs. 20.0 mm; *p* < 0.05), general VAS (38.0 mm vs. 26.0 mm; *p* < 0.05), and JSSF RA scale (70.0 points vs. 88.5 points; *p* < 0.001) improved significantly postoperatively. Hallux valgus angle (32.5° vs. 15.8°; *p* < 0.001), the angle comprising the first metatarsal and the second metatarsal (13.6° vs. 9.7°; *p* < 0.001) and the angle comprising the first metatarsal and the fifth metatarsal (35.8° vs. 25.7°; *p* < 0.001) decreased significantly one year after surgery (Table 2).

Preoperative and postoperative peak pressure distribution is shown in Figure 2. Peak pressure increased significantly at the first MTPJ (0.045 kg/cm^2^ vs. 0.082 kg/cm^2^; *p* < 0.05) and decreased significantly at the second and third MTPJs (0.081 kg/cm^2^ vs. 0.048 kg/cm^2^; *p* < 0.05, 0.097 kg/cm^2^ vs. 0.054 kg/cm^2^; *p* < 0.05). No significant differences in peak pressure were observed at the other sections: the first IPJ (0.045 kg/cm^2^ vs. 0.046 kg/cm^2^; *p*: 0.62), the fourth MTPJ (0.048 kg/cm^2^ vs. 0.036 kg/cm^2^; *p*: 0.06), the fifth MTPJ (0.032 kg/cm^2^ vs. 0.020 kg/cm^2^; *p*: 0.14), the medial midfoot (0.015 kg/cm^2^ vs. 0.016 kg/cm^2^; *p*: 0.30), the lateral midfoot (0.033 kg/cm^2^ vs. 0.033 kg/cm^2^; *p*: 0.17), and the hindfoot (0.050 kg/cm^2^ vs. 0.049 kg/cm^2^; *p*: 0.81). Preoperative and postoperative anteroposterior radiographs and plantar pressure distribution images of a representative case are shown in Figure 3. The pressure increased at the first MTPJ (0.008 kg/cm^2^ vs. 0.094 kg/cm^2^) and decreased at the second and third MTPJs (0.156 kg/cm^2^ vs. 0.092 kg/cm^2^, 0.165 kg/cm^2^ vs. 0.076 kg/cm^2^).

The percentages in which a maximum peak pressure value or a minimum peak pressure value occurred in each of the nine sections is shown in Table 3. A preoperative maximum peak pressure value was observed in 23.1% of feet at the first MTPJ and the third MTPJ, and in 19.2% of feet at the second MTPJ and the hindfoot. A postoperative maximum peak pressure was observed in 34.6% of feet at the first MTPJ and in 23.1% of feet at the first IPJ. A preoperative minimum peak pressure value was observed in 30.8% of feet at the first MTPJ and the fifth MTPJ, and in 26.9% of feet at the medial midfoot. A postoperative minimum peak pressure value was observed in 46.2% of feet at the medial midfoot and in 30.8% of feet at the fifth MTPJ.

## 4. Discussion

Mitchell’s osteotomy and its modified version have been reported in numerous studies dating from 1958 [21]. Both are relatively simple and effective reconstructive procedures for correcting mild to moderate hallux valgus. Transfer metatarsalgia has been seen as the most common complication of mMO [30]. It has been reported that osteotomy of the lesser metatarsals combined with osteotomy of the first metatarsal for hallux valgus correction helps prevent transfer metatarsalgia after mMO [22,31]. However, these reports are based solely on clinical assessment. Conversely, it has been reported that mMO alone does not result in a higher rate of residual metatarsalgia compared with the combination of osteotomies [32]. The present study, which evaluates changes in plantar pressure distribution following mMO and SOO, is a meaningful study in support of the suggestion that such a combination helps prevent transfer metatarsalgia.

There have been few studies investigating plantar pressure distribution after Mitchell’s osteotomy, or mMO [33]. Dhukaram et al. measured plantar pressure distribution after Mitchell’s osteotomy and demonstrated that deficient load bearing at the hallux and overloading at the second and third metatarsal heads were present [23]. Conversely, in the present study, although mMO was performed for the first metatarsal neck, postoperative peak pressure was found to be significantly decreased at the second and third MTPJs (Figure 2). The results of the present study suggest that a combination of mMO and SOO is a suitable one. In addition, unlike the previous studies, another important feature of the present study was that the subjects were patients with RA, not forefoot deformity alone.

Previous studies have reported on plantar pressure changes in patients with RA following other joint-preserving surgical techniques: Ebina et al. reported that a significant increase in plantar pressure at the first MTPJ and a significant decrease in plantar pressure at the second-third MTPJ occurred in RA patients following Scarf osteotomy and off-set osteotomy [16]; Shimoda et al. also reported a significant decrease in plantar pressure at the second and the third–fifth MTPJs and a significant increase in plantar pressure at the second toe following modified Scarf osteotomy and SOO [10]. However, their study also included a number of patients who underwent silicon joint replacement or resection arthroplasty [10]. Results similar to those found in the aforementioned studies were found in the present study. Despite the differences in surgical procedure, the procedures of the previous studies share a fundamental similarity to those of the present study, namely that correction of the rheumatoid forefoot deformity was achieved through shortening at the first through fifth metatarsals. This appears to be a key point for achieving optimal plantar pressure distribution in patients with RA. Among the various surgical procedures and combinations reported to date, the procedures used in the present study offer not only an effective combination, but also one with a greater degree of simplicity in that mMO and SOO are easier to perform than Scarf and off-set osteotomy, respectively. However, the radiographic parameters outlined in the report from Ebina et al. are superior to those of the present study. Therefore, their surgical procedure might be more suitable for RA patients with severe joint deformity in the first MTPJ.

A previous study of ours reported a significant decrease in peak pressure at the first IPJ and a significant increase in peak pressure at the midfoot following a surgical procedure in which silicon joint replacement for the first MTPJ was performed in conjunction with SOO for the second through fifth metatarsal necks [17]. Conversely, the combined surgical procedure in the present study resulted in no significant change in peak pressure at either the first IPJ or the midfoot (Figure 2). In addition, the percentage of feet in which a maximum peak pressure value occurred at the first IPJ changed from 7.7% preoperatively to 23.1% postoperatively, and the percentage of feet in which a minimum peak pressure value occurred at the medial midfoot changed from 26.9% preoperatively to 46.2% postoperatively (Table 3). The changes in the percentages observed at the first IPJ and the medial midfoot in the present study seem to reflect the findings of a study conducted by Yano et al., in which movement of the center of foot pressure distally was reported following joint-preserving surgery [18]. Taken together, these results indicate that the preservation of the first MTPJ helps to maintain the force generated by the extension of the first toe with the strain of the plantar aponeurosis. Elevated peak pressure at the midfoot section has been reported to be associated with falls in patients with RA [34]. Therefore, an additional benefit of this combined surgical procedure could be the reduced risk of falls.

The features of the patients in the present study are as follows: Preoperative forefoot deformation was 32.5 ° for hallux valgus angle and 13.6 ° for the angle comprising the first and second metatarsal, which was not particularly severe (Table 2). In addition, disease activity was seen to be low before surgery (Table 2). Preoperative disease activity in the objectives was 2.6 for the DAS28-ESR and 9.2 for the SDAI. Preoperative physical function was 0.4 for the J-HAQ, which indicates functional remission (Table 2). Meanwhile, among the objectives, pain and general VAS and JSSF RA scale improved significantly (Table 2). These results seem to reflect the decrease of postoperative peak pressure which was observed at the second and third MTPJs. Although MTX, biologics, and PSL usage ratios increased slightly one year after surgery, significant differences were not noted between the values recorded before surgery and one year after surgery (Table 2). Therefore, it is likely that the improvement can be attributed mainly to the surgery. The combination procedure of mMO and SOO used in the present study appears to be an effective and beneficial option for RA patients with mild to moderate joint deformity in the first MTPJ. Many such patients wish to maintain a healthy lifestyle in which activities requiring force generated by the extension of the first toe can be performed without hindrance. The recent remarkable advances in medication for RA have made it possible for RA patients to enjoy a greater degree of physical activity in their daily lives [35]. We are confident that the combination procedure described in the present study can meet the needs of such patients.

There are some limitations to the present study. First, the sample size was relatively small. The calculated sample size was 22 for the Wilcoxon’s signed rank test (G*Power3 software; test family: t tests; statistical test: Wilcoxon’s signed rank test; type of power analysis: A priori). Therefore, while small, the sample size in the present study was deemed sufficient for statistical analysis. Second, the follow-up period was short. Plantar pressure distribution has been reported to take about one year to stabilize [36]. Therefore, while short, the follow-up period in the present study occurred within the minimal timeframe. However, due to the chronic nature of RA, more research with longer-term follow-up needs to be conducted. Third, patient satisfaction was not evaluated in the present study. However, in addition to pain VAS, general VAS also decreased significantly one year after surgery. Therefore, we can assume that an assessment of patient satisfaction would be favorable one year after surgery. Fourth, the patients in the present study were all Japanese. It is possible that different plantar pressure distribution exists in different ethnicities. In the future, more research involving subjects from other ethnic groups should be conducted.

## 5. Conclusions

In conclusion, clinical background parameters improved significantly following mMO and SOO. Peak pressure increased significantly at the first MTPJ and decreased significantly at the second and third MTPJs. Based on these results, the combination of mMO and SOO can be an effective and beneficial option for RA patients with mild to moderate joint deformity. Future research on plantar pressure distribution is necessary to reveal which procedures and which combinations are the most optimal in order to achieve and maintain the functional remission of RA.

## Figures and Tables

**Figure 1 ijerph-18-09948-f001:**
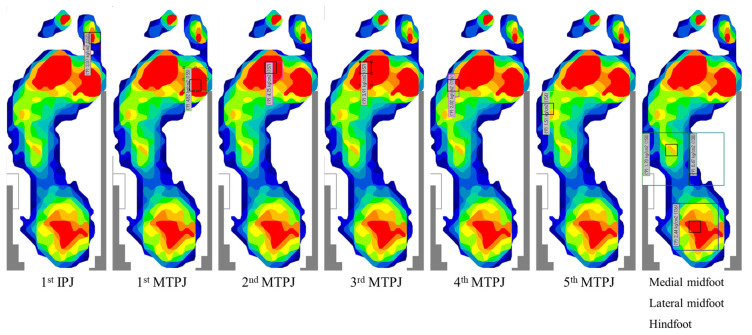
Plantar pressure measurement: IPJ = interphalangeal joint; MTPJ = metatarsophalangeal joint. Peak pressure was calculated at the following nine sections of interest, identified using radiography: the first IPJ, the first, second, third, fourth and fifth MTPJs, the medial midfoot, the lateral midfoot, and the hindfoot.

**Figure 2 ijerph-18-09948-f002:**
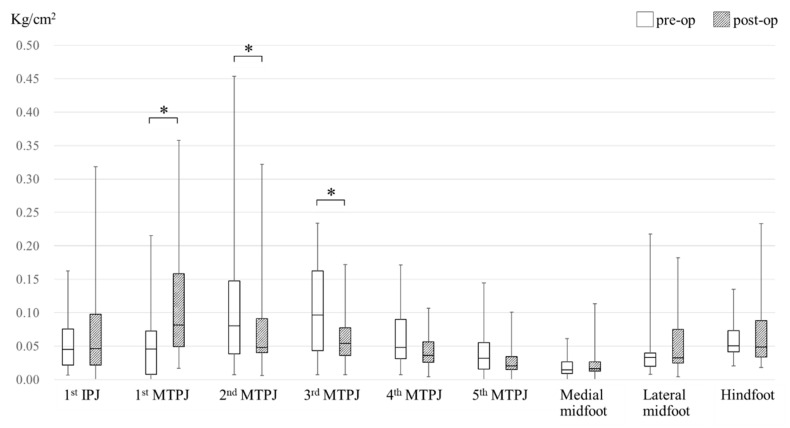
Preoperative and postoperative median peak pressure value. IPJ = interphalangeal joint; MTPJ = metatarsophalangeal joint. A significant difference between preoperative and postoperative values is indicated by an asterisk. *: *p* < 0.05.

**Figure 3 ijerph-18-09948-f003:**
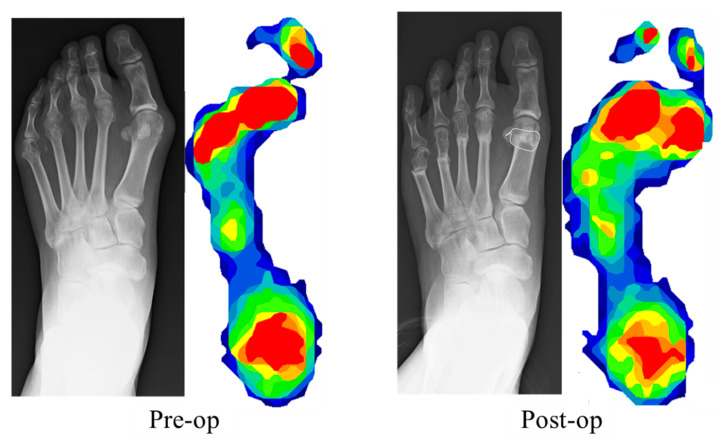
Preoperative and postoperative anteroposterior radiographs and plantar pressure distribution images of a representative case. Peak pressure increased at the first metatarsophalangeal joint (0.008 kg/cm^2^ vs. 0.094 kg/cm^2^) and decreased at the second and third metatarsophalangeal joints (0.156 kg/cm^2^ vs. 0.092 kg/cm^2^, 0.165 kg/cm^2^ vs. 0.076 kg/cm^2^).

**Table 1 ijerph-18-09948-t001:** Patient characteristics.

Number of patients	23
Number of female (%)	21 (91.3)
Number of feet	26
Age, years	60.0 (48.0–70.0)
Disease duration of RA, years	11.0 (7.0–15.0)
Body mass index, kg/m^2^	21.5 (19.9–23.3)
Larsen grade for the first MTPJ (number)	II:16, III:10

RA = rheumatoid arthritis, MTPJ = metatarsophalangeal joint. All values are the median and interquartile range, except when indicated otherwise.

**Table 2 ijerph-18-09948-t002:** Patient background parameters.

	Preoperatively	One Year Potoperatively
Methotrexate usage, number (%)	17 (73.9)	19 (82.6)
Biologics usage, number (%)	9 (39.1)	11 (47.8)
Prednisolone usage, number (%)	13 (56.5)	15 (65.2)
Pain visual analog scale, mm	31.0 (10.0–50.3)	20.0 (6.5–40.0) *
General visual analog scale, mm	38.0 (19.0–60.3)	26.0 (8.0–46.5) *
DAS28-ESR	2.6 (2.2–3.4)	2.7 (2.0–3.5)
SDAI	9.2 (5.8–12.0)	7.6 (2.5–10.0)
J-HAQ	0.4 (0–0.6)	0.4 (0–0.8)
JSSF RA scale		
General pain, 0–30 points	20.0 (20.0–30.0)	30.0 (30.0–30.0) **
Deformity, 0–25 points	12.0 (10.0–17.5)	21.5 (16.5–24.5) ***
Motion, 0–15 points	13.0 (13.0–15.0)	13.0 (13.0–13.0)
Walking ability, 0–20 points	20.0 (20.0–20.0)	20.0 (20.0–20.0)
Activities of daily living, 0–10 points	5.0 (3.0–7.0)	6.0 (4.0–7.0)
Total, 0–100 points	70.0 (67.0–80.5)	88.5 (85.0–91.0) ***
HVA, degree	32.5 (27.3–42.1)	15.8 (12.2–19.5) ***
M1/2, degree	13.6 (11.6–15.9)	9.7 (8.1–11.8) ***
M1/5, degree	35.8 (33.3–39.5)	25.7 (21.8–27.8) ***

DAS28-ESR = disease activity score in 28 joints based on erythrocyte sedimentation rate; SDAI = simplified disease activity index; J-HAQ = Japanese version of the Stanford health assessment questionnaire; JSSF RA scale = the Japanese Society for Surgery of the foot rheumatoid arthritis foot and ankle scale; HVA = hallux valgus angle; M1/2 = angle comprising the first and second metatarsal; M1/5 = angle comprising the first and fifth metatarsal. Except where indicated otherwise, values are the median and interquartile range. A significant difference between preoperative and postoperative values is indicated by an asterisk. *: *p* < 0.05, **: *p* < 0.01, ***: *p* < 0.001.

**Table 3 ijerph-18-09948-t003:** Percentage in which a maximum peak pressure value or a minimum peak pressure value occurred in each of the nine sections.

(a). Maximum Peak Pressure										
		First IPJ	First MTPJ	Second MTPJ	Third MTPJ	Fourth MTPJ	Fifth MTPJ	Medial Midfoot	Lateral Midfoot	Hind-foot
Pre-op (*n* = 26)	% (*n*)	7.7 (2)	23.1 (6)	19.2 (5)	23.1 (6)	3.8 (1)	3.8 (1)	0 (0)	0 (0)	19.2 (5)
Post-op (*n* = 26)	% (*n*)	23.1 (6)	34.6 (9)	11.5 (3)	11.5 (3)	3.8 (1)	0 (0)	0 (0)	7.7 (2)	7.7 (2)
**(b). Minimum Peak Pressure**										
		**First IPJ**	**First MTPJ**	**Second MTPJ**	**Third MTPJ**	**Fourth MTPJ**	**Fifth MTPJ**	**Medial Midfoot**	**Lateral Midfoot**	**Hind-foot**
Pre-op (*n* = 26)	% (*n*)	3.8 (1)	30.8 (8)	0 (0)	0 (0)	0 (0)	30.8 (8)	26.9 (7)	7.7 (2)	0 (0)
Post-op (*n* = 26)	% (*n*)	11.5 (3)	0 (0)	0 (0)	3.8 (1)	0 (0)	30.8 (8)	46.2 (12)	7.7 (2)	0 (0)

IPJ = interphalangeal joint; MTPJ = metatarsophalangeal joint. Values are presented as percent or number.

## Data Availability

The data presented in this study are available on request from the corresponding author. The data are not publicly available due to our institutional policy.

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
