# Peer review of "The Combination of Modified Mitchell’s Osteotomy and Shortening Oblique Osteotomy for Patients with Rheumatoid Arthritis: An Analysis of Changes in Plantar Pressure Distribution"

_ijerph, 2021, doi:10.3390/ijerph18199948_

Round 1
Reviewer 1 Report
Although the work presented has a low sample of subjects studied, the results are interesting to consider its publication in ijerhp, but first the authors must reflect on:
The authors describe in the abstract and methodology that the work consists of a reprospective study, but reading the manuscript concluded that it is prospective because the data were collected before and after the intervention, to be reprospective the authors would have collected the post-treatment measurements and would have searched for the pre-treatment data afterwards. This should be clarified.
The authors makea good introduction and justification of the work, but I recommend that the objective be reformulated more clearly and directly.
In line 89 it is stated that the subjects must have calluses on the forefoot, the authors must detail more the location of the callus, a collosity on the back of a finger is not the same as under the heads of the metatarsals ... both locations are in the forefoot.
The manuscript does not detail whether the subjects accepted their participation in the study, it would not be enough with the consent to be intervened, they are different issues.
Congratulations to the authors for the detailed description of the surgical interventions, but I imagine that these techniques were previously described so they must be endorsed with bibliography.
Authors should detail what pain visual analog scale, general visual analog scale consist of, or add bibliographic citations that describe them
The authors must include some image taken with the F-SCAN II (Nitta Corporation, Osaka, Japan) and how the measurements were made, to facilitate the understanding of section 2.4. and observe the pre and post results visually.
Reviewer 2 Report
The authors have done a fine job, but some changes need to make it a great contribution to this literature.
- Abstract:
The description of the following procedure, “surgical combination: modified Mitchell’s osteotomy performed for the first metatarsal neck and shortening oblique osteotomy performed for the lateral four metatarsal necks” should be simplified.
There is no result to support the following sentence, “transfer metatarsalgia might occur at the lateral MTPJs following modified Mitchell’s osteotomy”, and instead please show the numbers and p-value, especially in the results of plantar pressure distribution.
- Introduction:
Line 32; What do the author mean “lessor toe” in this sentence? Please modify “lessor toe” to “lesser toe” in this manuscript.
Line 52-53, Line 55-56; Please describe the differences and the reduction in plantar pressure distribution more detail.
Line 56-59; Ref.16 also showed the plantar peak pressure distribution after a joint-preserving arthroplasty, so the results should be added in the introduction and discussed the difference from previous studies in the discussion section.
- Methods:
In the “study and participants“ section, Larsen grade should be used to assess not deformity of the MTPJ but the severity of destruction of that.
In the “Surgical procedure” section, please describe the postoperative weight bearing.
In the “Plantar pressure measurement” section, please describe clearly that the maximum and minimum peak pressure indicates the highest and lowest peak pressure value found among all nine sections.
- Results
The abbreviations should be used for all following words, “methotrexate, prednisolone, pain visual analog scale, general visual analog scale, disease activity score in 28 joints based on erythrocyte sedimentation rate, simplified disease activity index, health assessment questionnaire disability index, the Japanese Society for Surgery of the Foot RA foot and ankle scale”.
This section doesn’t have very many numbers in it and the description of the results are not clear. Please include the numbers, especially in the results of peak pressure distribution, and describe all results more detail.
In Table2, please explain the ‘Lower HAQ’ and show a reference of this in the methods section. In addition, the place of JSSF scale should be modified.
In Table3a, please modify “number”.
The pre- and postoperative radiographs and images of plantar pressure distribution should be included as Ref.17 has shown.
- Discussion
There have been several papers investigated the plantar pressure distribution after the joint-preserving surgeries. This paper focused on the results of Mitchell’s osteotomy with shortening oblique osteotomy of the lesser toe but the difference of plantar pressure distribution from other joint-preserving surgeries should be discussed.
The description of the following procedure, “The combination procedure used in the present study in which modified Mitchell’s osteotomy is performed for the first metatarsal neck and shortening oblique osteotomy is performed for the second through fifth metatarsal necks” should be simplified.
Round 2
Reviewer 2 Report
The manuscript has been improved.
In Table2, please use the abbreviations for all following words, “methotrexate, prednisolone, pain visual analog scale, general visual analog scale.
This manuscript is a resubmission of an earlier submission. The following is a list of the peer review reports and author responses from that submission.